# Urban Driver: Learning to Drive from Real-world Demonstrations Using Policy Gradients

**Oliver Scheel, Luca Bergamini, Maciej Wołczyk, Błażej Osiński, Peter Ondruska**
Woven Planet, Level 5
{firstname.lastname}@woven-planet.global

**Abstract:** In this work we are the first to present an offline policy gradient method for learning imitative policies for complex urban driving from a large corpus of real-world demonstrations. This is achieved by building a differentiable data-driven simulator on top of perception outputs and high-fidelity HD maps of the area. It allows us to synthesize new driving experiences from existing demonstrations using mid-level representations. Using this simulator we then train a policy network in closed-loop employing policy gradients. We train our proposed method on 100 hours of expert demonstrations on urban roads and show that it learns complex driving policies that generalize well and can perform a variety of driving maneuvers. We demonstrate this in simulation as well as deploy our model to self-driving vehicles in the real-world. Our method outperforms previously demonstrated state-of-the-art for urban driving scenarios – all this without the need for complex state perturbations or collecting additional on-policy data during training. We make code and data publicly available.

**Keywords:** Self-driving, Learning from Demonstrations, Planning, Simulation

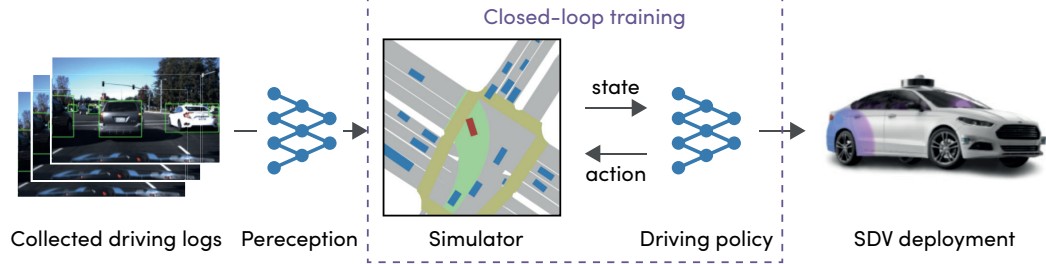

Figure 1: Overview of the proposed closed-loop training method for learning driving policies. We leverage large amounts of expert demonstrations and mid-to-mid representations to build a differentiable simulator supporting fast policy learning. With this simulator, we can effectively unroll model policies, and thus train the model closed-loop using a policy gradient method.

## 1 Introduction

Self-driving has the potential to revolutionize transportation and is a major field of AI applications. Even though already in 1990 there were prototypes capable of driving on highways [1], technology is still not widespread, especially in the context of urban driving. In the past decade, the availability of large datasets and high-capacity neural networks has enabled significant progress in perception [2, 3] and the vehicles' ability to understand their surrounding environment. Self-driving decision making, however, has seen very little benefit from machine learning or large datasets. State-of-the-art planning systems used in industry [4] still heavily rely on trajectory optimisation techniques with expert-defined cost functions. These cost functions capture desirable properties of the future vehicle path. However, engineering these cost functions scales poorly with the complexity of driving situations and the long tail of rare events.

Due to this, learning a driving policy directly from expert demonstrations is appealing, since performance scales to new domains by adding data rather than via additional human engineering effort.

5th Conference on Robot Learning (CoRL 2021), London, UK.

In this paper we focus specifically on learning rich driving policies for urban driving from large amounts of real-world collected data. Unlike highway driving [5], urban driving requires performing a variety of maneuvers and interactions with, e.g., traffic lights, other cars and pedestrians.

Recently, rich mid-level representations powered by large-scale datasets [6, 7], HD-maps and high-performance perception systems enabled capturing nuances of urban driving. This led to new methods achieving high performance for motion prediction [8, 9]. Furthermore, [10] demonstrated that leveraging these representations and behavioral cloning with state perturbations leads to learning robust driving policies. While promising, difficulty of this approach lies in engineering the perturbation noise mechanism required to avoid covariate shift between training and testing distribution.

Inspired by this approach, we present the first results on offline learning of imitating driving policies using mid-level representations, a closed-loop simulator and a policy gradient method. This formulation has several benefits: it can successfully learn high-complexity maneuvers without the need for perturbations, implicitly avoid the problem of covariate shift, and directly optimize imitation as well as auxiliary costs. The proposed simulator is constructed directly from the collected logs of real-world demonstrations and HD maps of the area, and can synthesize new realistic driving episodes from past experiences (see Figure 1 for an overview of our method). Furthermore, for training on large datasets reducing the computational complexity is paramount. We leverage vectorized representations and show how this allows for computing policy gradients quickly using backpropagation through time. We demonstrate how these choices lead to superior performance of our method over the existing state-of-the-art in imitation learning for real-world self-driving planning in urban areas.

Our contributions are four-fold:

- The first demonstration of policy gradient learning of imitative driving policies for complex urban driving from a large corpus of real-world demonstrations. We leverage a closed-loop simulator and rich, mid-level vectorized representations to learn policies capable of performing a variety of maneuvers.

- A new differentiable simulator that enables efficient closed-loop simulation of realistic driving experiences based on past demonstrations, and quickly compute policy gradients by backpropagation through time, allowing fast learning.

- A comprehensive qualitative and quantitative evaluation of the method and its performance compared to existing state-of-the-art. We show that our approach, trained purely in simulation can control a real-world self-driving vehicle, outperforms other methods, generalizes well and can effectively optimize both imitation and auxiliary costs.

- Source code and data are made available to the public[1].

## 2   Related Work

In this section we summarize different approaches for solving self-driving vehicle (SDV) decision-making in both academia and industry. In particular, we focus on both optimisation-based and ML-based systems. Furthermore, we discuss the role of representations and datasets as enablers in recent years, to tackle progressively more complex Autonomous Driving (AD) scenarios.

**Trajectory-based optimization** is still a dominant approach used in industry for both highway and urban-driving scenarios. It relies on manually defined costs and reward functions that describe good driving behavior. Such cost can then be optimized using a set of classical optimization techniques (A* [11], RRTs [12], POMDP with solver [13], or dynamic programming [14]). Appealing properties of these approaches are their interpretability and functional guarantees, which are important for AD safety. These methods, however, are very difficult to scale. They rely on human engineering rather than on data to specify functionality. This becomes especially apparent when tackling complex urban driving scenarios, which we address in this work.

**Reinforcement learning (RL)** [15] removes some complexity of human engineering by providing a reward (cost) signal and uses ML to learn an optimal policy to maximize it. Directly providing the reward through real-time disengagements [16], however, is impractical due to a low sample-efficiency of RL and the involved risks. Therefore, most approaches [17] rely on constructing a

---

[1]Code and video available at https://planning.l5kit.org.

simulator and explicitly encoding and optimising a reward signal [18]. A limiting factor of these approaches is that the simulator often is hand-engineered [19, 20], limiting its ability to capture long-tail real-world scenarios. Recent examples of sim-to-real policy transfer (e.g. [21], [22], [23]) were not focused on evaluating scenarios typical to urban driving, in particular interacting with other agents. In our work, we construct the simulator directly from real-world logs through mid-level representations. This allows training in a variety of real-world scenarios with other agents present, while employing efficient policy-gradient learning.

**Imitation learning (IL) and Inverse Reinforcement Learning (IRL)** [24, 25] are more scalable ML approaches that leverage expert demonstrations. Instead of learning from negative events, aim is to directly copy expert behavior or recover the underlying expert costs. Simple behavioral cloning was applied already back in 1989 [26] on rural roads, more recently by [27] on highways and [28] in urban driving. Naive behavioral cloning, however, suffers from covariate shift [24]. This issue has been successfully tackled for highway lane-following scenarios by reframing the problem as classification task [29] or employing a simple simulator [5], constructed from highway cameras. We take inspiration from these approaches but focus on the significantly more complex task of urban driving. Theoretically, our work is motivated by [30], as we employ a similar principle of generating synthetic corrections to simulate querying an expert. Due to this, identical proven guarantees hold for our method, namely the ideal linear regret bound, mitigating the problem of covariate shift. Adversarial Imitation Learning comprises another important field [31, 32, 33], but, to the best of our knowledge, has seen little application to autonomous driving and no actual SDV deployment yet.

**Neural Motion Planners** are another approach used for autonomous driving. In [34] raw sensory input and HD-maps are used to estimate cost volumes of the goodness of possible future SDV positions. Based on these cost volumes, trajectories can be sampled and the lowest-cost one is selected to be executed. This was further improved in [35], where the dependency on HD-maps was dropped. To the best of our knowledge, these promising methods have not yet been demonstrated to drive a car in the real-world though.

**Mid-representations and the availability of large-scale real-world AD datasets** [6, 7] have been major enablers in recent years for tackling complex urban scenarios. Instead of learning policies directly from sensor data, the input of the model comprises the output of the perception system as well as an HD map of the area. This representation compactly captures the nuance of urban scenarios and allows large-scale training on hundreds or thousands of hours of real driving situations. This led to new state-of-the-art solutions for motion forecasting [8, 9]. Moreover, [10] demonstrated that using mid-representations, large-scale datasets and simple behavioral cloning with perturbations [36] can scale and learn robust planning policies. The difficulty of this approach, however, is in engineering the noise model to overcome the covariate shift problem. In our work we are inspired by this approach, but attempt to learn robust policies using policy gradient optimisation [37] featuring unrolling and evaluating the policy during training. This implicitly avoids the problem of covariate shift and leads to superior results. This approach is, however, more computationally expensive and requires a simulator. To solve this, we show how a fast and powerful simulator can be constructed directly from real-world logs enabling scalability of this approach.

**Data-driven simulation.** A realistic simulator is useful for both training and validation of ML models. However, many current simulators (e.g. [19, 38]) depend on heuristics for vehicle control and do not capture the diversity of real world behaviours. Data-driven simulators are designed to alleviate this problem. [23] created a photo-realistic simulator for training an end-to-end RL policy. [5] simulated a bird's-eye view of dense traffic on a highway. Finally, two recent works [39, 40] developed data-driven simulators and showed their usefulness for training and validating ML planners. In this work we show that a simpler, differentiable simulator based on replaying logs is effective for training.

## 3    Differentiable Traffic Simulator from Real-world Driving Data

In this section we describe a differentiable simulator $S$ that approximates new driving experiences based on an experience $\bar{\tau}$ collected in the real world. This simulator is used during policy learning for the closed-loop evaluation of the current policy's performance and computing the policy gradient. As shown in Section 5, differentiability is an important building block for achieving good results, especially when employing auxiliary costs.

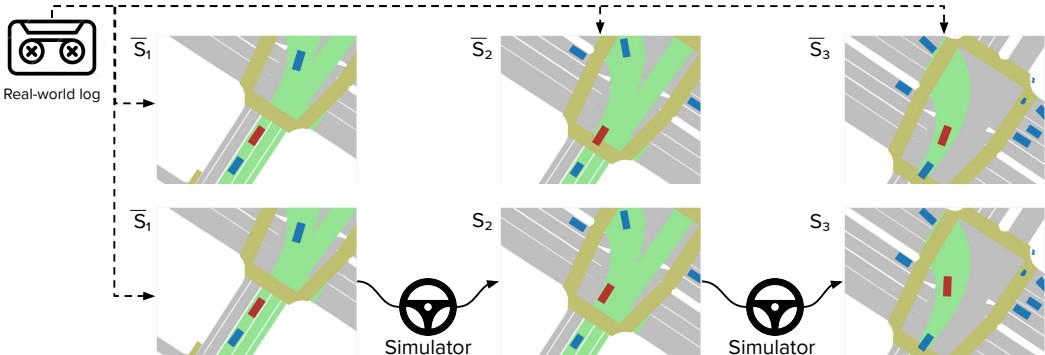

Figure 2: Differentiable simulator from observed real-world logs: based on a ground truth log (top row), we unroll a new trajectory corresponding to different SDV actions (e.g. given by a planner) in the simulator approximating the vectorized world representation (bottom row)

We represent the real-world experience $\bar{\tau}$ as a sequence of state observations $\bar{s}_t$ around the vehicle over time:

$$\bar{\tau} = \{\bar{s}_1, \bar{s}_2, ..., \bar{s}_T\}. \tag{1}$$

We use a vectorized representation based on [8], in which each state observation $\bar{s}_t$ consists of a collection of static and dynamic elements $e_t^1, e_t^2, ..., e_t^K$ around the vehicle pose $\bar{p}_t \in SE_2$, with $SE_2$ denoting the special Euclidean group. Static elements include traffic lanes, stop signs and pedestrian crossings. These elements are extracted from the underlying HD semantic map of the area using the localisation system. The dynamic elements include traffic lights status and traffic participants (other cars, buses, pedestrians and cyclists). These are detected in real-time using the on-board perception system. Each element $e_t^j$ includes a pose $q_t^j \in SE_2$ relative to the SDV pose $p_t$, as well as additional features, such as the element type, time of observation, and other optional attributes, e.g. the color of associated traffic lights, recent history of moving vehicles, etc. The full details of this representation are provided in Appendix C.

Goal of the simulation is to iteratively generate a sequence of state observations $\tau = \{s_1, s_2, \ldots, s_T\}$ that corresponds to a different sequence of driver actions $a_1, a_2, ..., a_N$ in the scenario. This is done by first computing the corresponding SDV trajectory $p_1, p_2, ..., p_N$ and then locally transforming states $\bar{s}_1, \bar{s}_2, \ldots, \bar{s}_N$.

Updated poses of the SDV are determined by a kinematic model $p_{t+1} = f(p_t, a_t)$, which is assumed to be differentiable. The state observation $s_t$ is then obtained by computing the new position $q_t^j$ for each state element $e_t^j$ using a transformation along the differences of the original and updated pose:

$$q_t^j = \bar{q}_t^j (p_t - \bar{p}_t). \tag{2}$$

See Figure 2 for an illustrative example. It is worth noting that this approximation is effective if the distance between the original and generated SDV pose is not too large.

We denote performing these steps in sequence with the step-by-step simulation transition function $s_{t+1} = S(s_t, a_t)$. Moreover, since both Equation (2) and vehicle dynamics $f$ are fully differentiable, we can compute gradients with respect to both the state ($S_s$) and action ($S_a$). This is critical for the efficient computation of policy gradients using backpropagation through time as described in the next section.

## 4   Imitation Learning Using a Differentiable Simulator

In this part, we detail how we use the simulator $S$ described in the previous section to learn a deterministic policy $\pi$ to drive a car using closed-loop policy learning.

We frame the imitation learning problem as minimisation of the L1 pose distance $L(s_t, a_t) = \|\bar{p}_t - p_t\|_1$ between the expert and learner on a sequence of collected real-world demonstrations $\bar{\tau}_1, \bar{\tau}_2, ..., \bar{\tau}_N \sim \pi_E$. Note that with a slight abuse of notation we use the poses $\bar{p}_t, p_t$ here to refer to 3D vectors $(x, y, \theta)$, instead of roto-translation matrices in $SE_2$ – yielding the common L1 norm

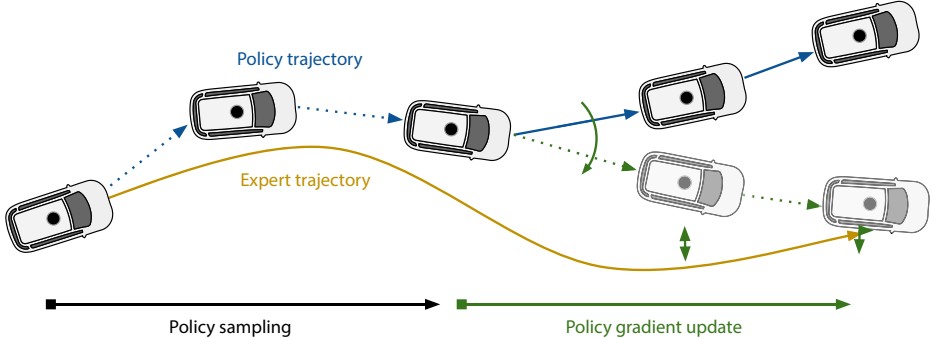

Figure 3: One iteration of policy gradient update. Given a real-world expert trajectory $\bar{\tau}$ we sample a policy state $s_t$ by unrolling the policy $\pi$ for $T$ steps. We then compute optimal policy update by backpropagation through time.

and loss. This can be expressed as a discounted cumulative expected loss [41] on the set of collected expert scenarios:

$$J(\pi) = \mathbb{E}_{\bar{\tau} \sim \pi_E} \mathbb{E}_{\tau \sim \pi} \sum_t \gamma^t L(s_t, a_t). \tag{3}$$

Optimising this objective pushes the trajectory taken by the learned policy as close as possible to the one of the expert, as well as limiting the trajectory to the region where the approximation given by the simulator is effective. In Appendix B we further extend this to include auxiliary cost functions with the aim of optimising additional objectives.

We can use any policy optimisation method [42, 43] to optimize Equation (3). However, given that the transition $S(s_t, a_t)$ is differentiable, we can exploit it for a more effective training that does not require a separate estimation of a value function. As shown in [31, 37, 44], this results into an order of magnitude more efficient training. The optimisation process consists of repeatedly sampling pairs of expert and policy trajectories $\bar{\tau}_i, \tau_i$ and computing the policy gradient $J_\theta$ for these samples to minimize Equation (3). We describe both steps in detail in the following subsections.

## 4.1 Sampling from a Policy Distribution $\pi$

In this subsection we detail sampling pairs of expert ($\bar{\tau}$) and corresponding policy trajectory ($\tau$) drawn from policy $\pi$.

Sampling expert trajectories $\bar{\tau}$ consists of simply sampling from the collected dataset of expert demonstrations. To generate the policy sample $\tau$ we acquire an expert state $\bar{s}_1 \in \bar{\tau}$, and then unroll the current policy $\pi$ for $T$ steps using the simulator $S$.

This naive method, however, introduces bias, as the initial state of the trajectory is always drawn from the expert $\pi_E$ and not from the policy distribution $\pi$. As shown in Appendix B, this results in the under-performance of the method. To remove this bias we discard the first $K$ timesteps from both trajectories and use only the remaining $T - K$ timesteps to estimate the policy gradient $J_\theta$ as described next (see Figure 3 for a visualization).

---

**Algorithm 1:** Imitation learning from expert demonstrations

**Input:** Expert policy samples
$$\bar{\tau}_1, \bar{\tau}_2, ..., \bar{\tau}_N \sim \pi_E$$
**Output:** Learned policy $\pi$
$\pi = \text{random}$ ;
**for** $\bar{\tau} \sim \pi_E$ **do**
    **for** $t = 1$ **to** $T$ **do**
        $a_t = \pi(s_t)$;
        $s_{t+1} = S(s_t, a_t)$;
    **end**
    $J_s^{T+1} = 0$;
    $J_\theta^{T+1} = 0$;
    **for** $t = T$ **downto** $K$ **do**
        $J_s^t = L_s + L_a \pi_s + \gamma J_\theta^{t+1}(S_s + S_a \pi_s)$;
        $J_\theta^t = L_a \pi_\theta + \gamma(J_s^{t+1} S_a \pi_\theta + J_\theta^{t+1})$;
    **end**
    $\pi = \text{gradient\_update}(\pi, J_\theta^K)$;
**end**

---

## 4.2 Computing Policy Gradient $J_\theta$

Here we describe the computation of the policy gradient $J_\theta$ around the rollout trajectory $\tau = s_1, a_1, s_2, a_2, \ldots, s_T, a_T$ given by the current policy. This gradient can be computed for deterministic policies $\pi$ using backpropagation through time leveraging the differentiability of the simulator $S$. Note that we denote partial differentiation with subscripts, i.e. $g_x \triangleq \partial g(x, \ldots)/\partial(x)$. We follow the formulation in [37] and express the gradient by a pair of recursive formulas:

$$J_s^t = L_s + L_a \pi_s + \gamma J_\theta^{t+1}(S_s + S_a \pi_s), \tag{4}$$

$$J_\theta^t = L_a \pi_\theta + \gamma(J_s^{t+1} S_a \pi_\theta + J_\theta^{t+1}). \tag{5}$$

The resulting algorithm is outlined in Algorithm 1 and illustrated in Figure 3. It can be implemented simply as one forward pass of length $T$ and one backward pass of length $T - K$. To compute the policy gradient we use equations (4) and (5) recursively from $t = T$ to $t = K$ and use it to update policy parameters $\theta$.

## 5 Experiments

In this section we evaluate our proposed method and benchmark it against existing state-of-the-art systems. In particular, we are interested in: its ability to learn robust policies dealing with various situations observed in the real world; its ability to tailor performance using auxiliary costs; the sensitivity of key hyper-parameters; and the impact on performance with increasing amounts of training data. Additional results can be found in the appendix and the accompanied video.

### 5.1 Dataset

For training and testing our models we use the Lyft Motion Prediction Dataset [6]. This dataset was recorded by a fleet of Autonomous Vehicles and contains samples of real-world driving on a complex, urban route in Palo Alto, California. The dataset captures various real-world situations, such as driving in multi-lane traffic, taking turns, interactions with vehicles at intersections, etc. Data was preprocessed by a perception system, yielding the precise position of nearby vehicles, cyclists and pedestrians over time. In addition, a high-definition map provides locations of lane markings, crosswalks and traffic lights. All models are trained on a 100h subset, and tested on 25h. The training dataset is identical to the publicly available one, whereas for the sake of execution speed for testing we use a random, but fixed, subset of the listed test dataset, which is roughly $\frac{1}{4}$ in size.

### 5.2 Baselines

We compare our proposed algorithm against three state-of-the-art baselines:

- Naive Behavioral Cloning (*BC*): we implement standard behavioral cloning using our vectorized backbone architecture. We do not use the SDV's history as an input to the model to avoid causal confusion (compare [10]).

- Behavioral Cloning + Perturbations (*BC-perturb*): we re-implement a vectorized version of ChauffeurNet [10] using our backbone network. As in the original paper, we add noise in the form of perturbations during training, but do not employ any auxiliary losses. We test two versions: without the SDV's history, and using the SDV's history equipped with history dropout.

- Multi-step Prediction (*MS Prediction*): we apply the meta-learning framework proposed in [30] to train our vectorized network. We observe that a version of this algorithm can conveniently be expressed within our framework; we obtain it by explicitly detaching gradients between steps (i.e. ignoring the full differentiability of our simulation environment). Differently from the original work [30], we do not save past unrolls as new dataset samples over time.

### 5.3 Implementation

Inspired by [8, 45], we use a graph neural network for parametrizing our policy. It combines a PointNet-like architecture for local inputs processing followed by an attention mechanism for global

| Configuration | | Collisions | | | Imitation | | Comfort | I1K |
|---|---|---|---|---|---|---|---|---|
| Model | SDV history | Front | Side | Rear | Off-road | L2 | | |
| BC | | $79 \pm 23$ | $395 \pm 170$ | $997 \pm 74$ | $1618 \pm 459$ | $1.57 \pm 0.27$ | $\mathbf{93K} \pm 3K$ | $3{,}091 \pm 601$ |
| BC-perturb | | $16 \pm 2$ | $56 \pm 6$ | $411 \pm 146$ | $82 \pm 11$ | $0.74 \pm 0.01$ | $203K \pm 6K$ | $567 \pm 128$ |
| BC-perturb | ✓ | $\mathbf{14} \pm 4$ | $73 \pm 7$ | $678 \pm 11$ | $\mathbf{77} \pm 6$ | $0.77 \pm 0.01$ | $636K \pm 22K$ | $843 \pm 6$ |
| MS Prediction | ✓ | $18 \pm 6$ | $55 \pm 4$ | $125 \pm 14$ | $141 \pm 31$ | $0.46 \pm 0.02$ | $595K \pm 49K$ | $341 \pm 39$ |
| Ours | ✓ | $15 \pm 7$ | $\mathbf{46} \pm 5$ | $\mathbf{101} \pm 13$ | $97 \pm 6$ | $\mathbf{0.42} \pm 0.00$ | $637K \pm 41K$ | $\mathbf{260} \pm 9$ |

Table 1: Normalized metrics for all baselines and our method – reporting mean and standard deviation for each as obtained from 3 runs. For all, lower is better. Our method overall yields best performance and lowest I1K.

reasoning. In contrast to [8], we use points instead of vectors. Given the set of points corresponding to each input element, we employ 3 PointNet layers to calculate a 128-dimensional feature descriptor. Subsequently, a single layer of scaled dot-product attention performs global feature aggregation, yielding the predicted trajectory. We found $K = 20$ and $T = 32$ to work well, i.e. we use 20 timesteps for the initial sampling and effectively predict 12 trajectory steps. $\gamma$ is set to 0.8. In total, our model contains around 3.5 million trainable parameters, and training takes 30h on 32 Tesla V100 GPUs. For more details we refer to Appendix C.

For the vehicle kinematics model $f$ we use an unconstrained model $p_{t+1} = p_t + a_t$ with $a_t \in SE_2$. This allows for a fair comparisons with the baselines as both BC-perturb and MS Prediction assume the possibility of arbitrary pose corrections. Other kinematics models, such as unicycle or bicycle models, could be used with our method as well.

All baseline methods share the same network backbone as ours, with model specific differences as described above – and BC and BC-perturb predicting a full T-step trajectory with a single forward, while MS Prediction and ours are calling the model $T$ times. To ensure a fair comparison, also for MS Prediction we use our proposed sampling procedure, i.e. use the first $K$ steps for sampling only. We train all models for 61 epochs with a learning rate of $10^{-4}$, and drop it to $10^{-5}$ after 54 epochs. We note that we achieve best results for our proposed method by disabling dropout, and hypothesize this is related to similar issues observed for RNNs [46].

We refer the reader to Appendix B for ablations on the influence on data and sampling.

## 5.4 Metrics

We implement the metrics describe below to evaluate the planning performance. These capture key imitation performance, safety and comfort. In particular, we report the following, which are normalized – if applicable – per 1000 miles driven by the respective planner:

- **L2**: L2 distance to the underlying expert position in the driving log in meters.
- **Off-road events**: we report a failure if the planner deviates more than 2m laterally from the reference trajectory – this captures events such as running off-road and into opposing traffic.
- **Collisions**: collisions of the SDV with any other agent, broken down into front, side and rear collisions w.r.t. the SDV.
- **Comfort**: we monitor the absolute value of acceleration, and raise a failure should this exceed 3 m/s$^2$.
- **I1K**: we accumulate safety-critical failures (collisions and off-road events) into one key metric for ease of comparison, namely *Interventions per 1000 Miles* (I1K).

## 5.5 Imitation Results

We evaluate our method and all the baselines by unrolling the policy on 3600 sequences of 25 seconds length from the test set and measure the above metrics.

Table 1 reports performance when all methods are trained to optimize the imitation loss alone. Behavioral cloning yields a high number of trajectory errors and collisions. This is expected, as this approach is known to suffer from the issue of covariate shift [24]. Including perturbation during training dramatically improves performance as it forces the method to learn how to recover from

drifting. We further observe that MS Prediction yields comparable results for many categories, while yielding less rear collisions. We attribute this to the further reduction of covariate shift when compared to the previous methods: the training distribution is generated on-policy instead of being synthesized by adding noise. Finally, our method yields best results overall. It is worth noting that all models share a high number of comfort failures, due to the fact that they are all trained for imitation performance alone, which does not optimize for comfort, but only positional accuracy of the driven vehicle – which we address in the appendix.

### 5.6 In-car Testing

In addition to above stated simulation results, we further deployed our planner on SDVs in the real. For this, a Ford Fusion equipped with 7 camera, 3 LiDAR and 11 Radar sensors was employed. The sensor setup thus equals the one used for data collection, and during road-testing our perception and data-processing stack is run in real-time to generate the desired scene representation on the fly. For this, vehicles are equipped with 8 Nvidia 2080 TIs. Experiments were conducted on a private test track, including other traffic participants and reproducing challenging driving scenarios. Furthermore, this track was never shown to the network before, and thus offers valuable insights into generalization performance. Figure 5 shows our model successfully crossing a signaled intersection, for more results we refer to the appendix and our supplementary video.

## 6 Conclusion

In this work we have introduced a method for learning an autonomous driving policy in an urban setting, using closed-loop training, mid-level representations with a data-driven simulator and a large corpus of real world demonstrations. We show this yields good generalization and performance for complex, urban driving. In particular, it can control a real-world self-driving vehicle, yielding better driving performance than other state-of-the-art ML methods.

We believe this approach can be further extended towards production-grade real-world driving requirements of L4 and L5 systems – in particular, for improving performance in novel or rarely seen scenarios and to increase sample efficiency, allowing further scaling to millions of hours of driving.

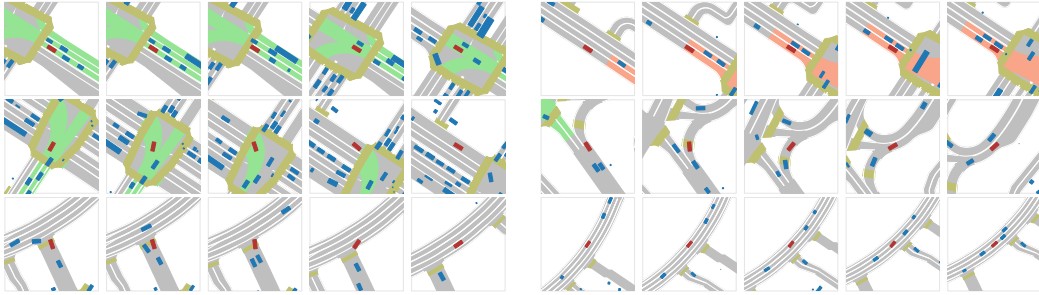

Figure 4: Qualitative results of our method controlling the SDV. Every row depicts two scenes, images are 2s apart. The SDV is drawn in red, other agents in blue and crosswalks in yellow. Traffic lights colors are projected onto the affected lanes. Best view on a screen.

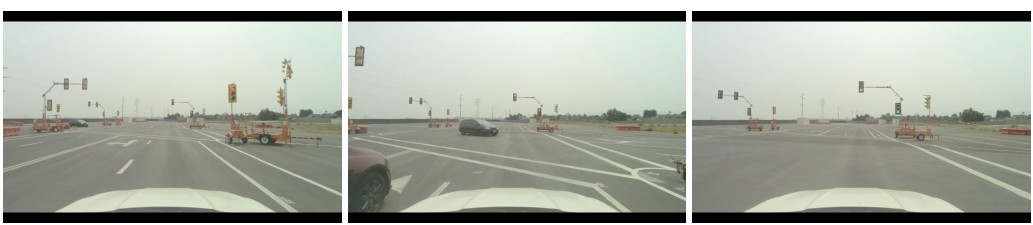

Figure 5: Front-camera footage of our planner stopping for a red light, and subsequently crossing the intersection when the signal turns green (images from left to right, recorded several seconds apart).

**Acknowledgments**

We would like to thank everyone at Level 5 working on data-driven planning, in particular Sergey Zagoruyko, Yawei Ye, Moritz Niendorf, Jasper Friedrichs, Li Huang, Qiangui Huang, Jared Wood, Yilun Chen, Ana Ferreira, Matt Vitelli, Christian Perone, Hugo Grimmett, Parth Kothari, Stefano Pini, Valentin Irimia and Ashesh Jain. Further we would like to thank Alex Ozark, Bernard Barcela, Alex Oh, Ervin Vugdalic, Kimlyn Huynh and Faraz Abdul Shaikh for deploying our planner to SDVs in the real.

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
