# OpenReview forum: "Urban Driver: Learning to Drive from Real-world Demonstrations Using Policy Gradients"
_robot-learning.org/CoRL/2021/Conference — CoRL2021 Poster_

### Official Review · Reviewer_oEVn · 2021-07-22

**Originality:** Good
**Technical Quality:** Fair
**Clarity Of Presentation:** Good
**Impact:** 3

**Recommendation:**

Weak Accept: I recommend accepting the paper, but will not argue for my recommendation if the majority of other reviewers have a different opinion.

**Summary:**

This paper constructs a simulation environment based on real traffic data and train a traffic agent by means of imitation learning. To alleviate the covariate shift phenomenon, this paper introduces the “Differentiable simulator” method. The authors compare the performances of their method with some classical methods of imitation learning and show that their method has better performance in some indicators such as collision rate. In addition, the paper also carries out comparative tests on some reinforcement learning kinds of auxiliary rewards.

**Issues:**

I think that this paper needs more empirical evidence that the proposed method outperforms other SOTA algorithm in imitation learning.

**Reviewer Expertise:**

Very good: Comprehensive knowledge of the area

**Strengths And Weaknesses:**

Strengths：
The first is the investigated problem is very meaningful for autonomous driving, which is how to imitate a human-like traffic agent policy in a complex traffic scenario from the real dataset.
The second is that this paper combines some current mainstream methods such as pointNet and differentiable forward loss for policy to improve the performance of the imitation learning.

The biggest weakness of the approach is, in my opinion, the lack of novelty. Indeed, all the core ideas of the contribution have been already proposed in existing methods. The idea of using differentiable simulator for reducing covariate shift was already proposed, with exactly the same loss style but different in the concrete form of calculation (this paper even use a more simplified form of Ls and La term, which in [35] are represent the backpropagation item towards policy parameter.) in [35] (in the paper). The use of reinforcement learning kinds of reward for helping revise the imitation learning policy behavior was already used in [1]. In addition, the baseline methods compared in this paper, One is behavior cloning inherently flawed in terms of covariate shift problem, which makes this comparison seem insufficiently persuasive. Therefore, the work appears to be very incremental at the moment, with only very small variations of existing methods.
[1] Bhattacharyya, R. , et al. "Modeling Human Driving Behavior through Generative Adversarial Imitation Learning." (2020).


**Summary Of Recommendation:**

Although the novelty is not enough, the investigated problem is very meaningful for autonomous driving. And the experimental result is convincing.

---

> ### Author Response · Authors · 2021-08-27
> **Addressing the raised issues**
>
> Dear Reviewer,
>
> Thank you very much for your helpful review and for taking interest in our work. We would kindly ask you to first read our general comments in the answer to the AC above, and in this comment address your points in particular.
>
> We thank you for your suggestion of including additional baselines. For our response and a statement e.g. concerning Adversarial Imitation Learning methods, please refer to the global response.
>
> We would argue though that our baselines already are strong, and - as stated in the principal answer above - reflect those methods which were actually proven to be scalable and also deployable in the real world, for the complex problem we are attempting, such as [1].
> Finally, we agree that the publication [2] is influential to our work and the community overall, but would also like to stress the difficulty of reproducibility [3] and above stated different focus, consequently missing the real-world deployment.
>
> -----------
> [1] “ChauffeurNet: Learning to Drive by Imitating the Best and Synthesizing the Worst”, Bansal, Krizhevsky, Ogale, 2018
> [2] “End-to-End Differentiable Adversarial Imitation Learning”, Baram, Anschel, Caspi, Mannor, 2017
> [3] “Sample efficient imitation learning for continuous control.” Sasaki, Yohira, Kawaguchi. ICLR 2018. - in this paper, on Page 7 in the footnote the authors mentioned that they could not reproduce MGAIL's results.

---

### Official Review · Reviewer_KT1F · 2021-07-24

**Originality:** Fair
**Technical Quality:** Fair
**Clarity Of Presentation:** Good
**Impact:** 3

**Recommendation:**

Weak Accept: I recommend accepting the paper, but will not argue for my recommendation if the majority of other reviewers have a different opinion.

**Summary:**

The paper proposes a simulation-based, imitation learning-based approach for learning policies for urban driving scenarios from real-world human driving trajectories. To train the policies, a simulator is developed to replay traffic and simulate the ego vehicle, where the ego vehicle action is governed by a policy model utilizing PointNet and Multi-Head Attention layers. The policy model takes HD map, perceived objects' polygons, and the ego vehicle polygon as inputs and outputs a series of position coordinates and orientations as a trajectory. The model is trained by minimizing the error between the generated trajectories and those of experts. The results are compared with behavioral cloning and multi-step prediction methods.

**Issues:**

1. The comfort method is based on longitudinal acceleration. It's recommended to consider jerk and/or lateral acceleration instead as they are more representative of comfort in driving.
2. An off-road event threshold seems to be too large (4 m). Considering that lane width ranges between 3-4 m and vehicles usually drive in the center of lanes, the threshold could be reduced to 1.5-2 m to capture off-lane events.
3. Referring to Table 1, BC and MS Prediction results seem to be very close to that of the proposed method when considering front and side collisions, but significantly higher for rear collisions. The reason behind this needs to be elaborated on in the paper. Should rear collisions be considered a failure?
4. The comparison results can be extended to more recent works including IRL/IM papers mentioned in the related work section.
5. There is no result to judge if the learned policy can generalize to different maps or scenarios. Further tests need to be provided to check the performance of the policy in a novel, but a similar map.
6. The simulator differentiably needs to be discussed in the case of collision, sudden appearance/disappearance of objects.
7. The related works need more recent methods to cover the whole problem including other end-to-end methods.
8. The difference between the presented simulator and the differentiable simulator presented in [1] needs to be highlighted.
9. Line 165-167, "To remove this bias we discard the first K timesteps ...", an experiment showing a comparison between with or without discarding K timesteps is helpful.
10. The introduction discusses covariant shifts between training and test distributions, but it is not clear how the proposed approach has addressed this problem.

**Reviewer Expertise:**

Excellent: Expert knowledge on the topic of the paper

**Strengths And Weaknesses:**

The main strength of the paper is on presenting a differentiable traffic simulator from real-world driving data, which can potentially be used to generate synthesized driving scenarios and as a testbed for autonomous driving applications.  However, the paper does not provide sufficient details on the simulator and how the traffic is managed in the simulator. It seems that the traffic is simulated simply by replaying data. This could be problematic in many autonomous driving scenarios including scenarios, in which the ego vehicle behavior is slightly different from the expert demonstration.  For example, consider a driving case where the speed of the ego vehicle is marginally lower than the expert driver when stopping at a red traffic light. Without adjusting the traffic, this could result in a rear-end collision. This can become more critical when there is a significant difference between the policy and demonstration. For example, consider a case where the policy decides to make a valid lane change instead of remaining within the lane to follow the demonstration.  Given these cases, the application of the simulator seems to be restricted without considering a proper traffic management system. Additionally, the simulator's differentiablity seems to be narrowed down to the kinematic model of the ego vehicle, which is further reduced to a simple motion model. The simulator is not detailed adequately to judge whether perception inaccuracy or collision can lead to discontinuity or not. For example, does the simulator remains differentiable in case of collision or disappearance of an object? Lastly, it's not clear if the result is generalizable to different maps or scenarios.

**Summary Of Recommendation:**

The simulator is not detailed enough to judge its usability and impact on autonomous driving research and if there are any differences between the presented simulator and other simulators (e.g. [1]). The imitative policy seems to be just another model without being properly validated through ablation tests, generalizability results, and sufficient comparison results with SOTA. While less important, the provided comparison results also do not show a significant/clear improvement to BC.

---

> ### Author Response · Authors · 2021-08-17
> **Missing reference**
>
> Dear Reviewer,
>
> Thank you very much for your very insightful and careful review of our work. As we are working on preparing answers to the raised concerns, I would like to ask about the article referenced in your review as [1]. It seems that the bibliographic information is missing in your review, would you mind letting us know which paper you are referring to?

---

> > ### Comment · Reviewer_KT1F · 2021-08-17
> > **Missing reference**
> >
> > Here is the missing reference:
> >
> > [1] Suo, S., Regalado, S., Casas, S., & Urtasun, R. (2021). TrafficSim: Learning to Simulate Realistic Multi-Agent Behaviors. In Proceedings of the IEEE/CVF Conference on Computer Vision and Pattern Recognition (pp. 10400-10409).

---

> ### Author Response · Authors · 2021-08-27
> **Addressing the raised issues, part 1**
>
> Dear Reviewer,
>
> Thank you very much for your insightful review and knowledgeable comments.
>
> We would kindly ask you to first read our general comments in the answer to the AC above, and in this comment address your points in particular.
>
> General concerns about the usage of log-replay, generalizability and ablation studies: We thank you for raising this point. We do see alternatives to log-replaying scenes in the simulator, but would like to point out the good results of our method - both in simulation as well as real-world deployment, the latter simultaneously proving generalizability of our model. We include different ablation studies in the appendix, and hope they are a valuable addition to the work and sheds further insights into our proposed method.
>
> > 1. The comfort method is based on longitudinal acceleration. It's recommended to consider jerk and/or lateral acceleration instead as they are more representative of comfort in driving.
>
> Main topic of this paper naturally is self-driving, however, we still aim to provide value for the full community, and also reach people working on - for example - Imitation Learning theory. Due to this we feel a single value representing comfort - namely acceleration - is easier to understand for the first part of the paper. However, in the appendix we repeat all tables and experiments with the metrics you described, namely (longitudinal) jerk and lateral acceleration. We thank you for this very insightful comment, and appreciate including metrics targeted at people with detailed domain knowledge.
>
> > 2. An off-road event threshold seems to be too large (4 m). Considering that lane width ranges between 3-4 m and vehicles usually drive in the center of lanes, the threshold could be reduced to 1.5-2 m to capture off-lane events.
>
> We thank you for this suggestion and have changed all results to respect a new error threshold of 2m, but also left the original results in the appendix to compare different thresholds - please note that this change also influences other metrics due to our way of evaluation (namely stopping / resetting after interventions). We agree that a threshold of 2m is reasonable to capture off-road events - note that our previous threshold was set somewhat higher to allow a slightly more relaxed evaluation, and we added a discussion on this in the appendix.
>
> > 3. Referring to Table 1, BC and MS Prediction results seem to be very close to that of the proposed method when considering front and side collisions, but significantly higher for rear collisions. The reason behind this needs to be elaborated on in the paper. Should rear collisions be considered a failure?
>
> Thank you for this important question, we have added a short discussion about this in Appendix B. You are right, that rear collisions often are caused by mistakes of the trailing agents or non-reactive simulation. Still, in our data we can identify grave misbehaviors of ego vehicle leading to rear collisions, which is why we include them in our summarized metric I1K. In Appendix B we show one such scene. However, note that due to this we took great care in sharing all our results and a full breakdown of metrics, s.t. every interested reader can work with the raw data, as well.
>
> > 4. The comparison results can be extended to more recent works including IRL/IM papers mentioned in the related work section.
>
> Please refer to the global answer for our justification of the selected baselines.
>
> > 5. There is no result to judge if the learned policy can generalize to different maps or scenarios. Further tests need to be provided to check the performance of the policy in a novel, but a similar map.
>
> We updated our supplementary video with our planner being deployed on an actual self-driving vehicle. These tests are executed on our testing ground, which represents a completely new and previously unseen environment for our ML planner - thus we argue this positively helps answer the question regarding generalization.

---

> > ### Author Response · Authors · 2021-08-27
> > **Addressing the raised issues, part 2**
> >
> > > 6. The simulator differentiably needs to be discussed in the case of collision, sudden appearance/disappearance of objects.
> >
> > Differentiability is not impeded through collisions, as our simulator does not provide a physics simulation of the world - collisions do not have an influence on unrolls. Further, please note that in Appendix C we list additional details describing our data processing pipeline - there explaining that data is queried one for each unroll, and we then execute the unroll in this world state. You are right to note that due to this, newly appearing objects are not accounted for - but consequently, also differentiability is still given. We reason such behavior with our assumption of “spatial and temporal closeness” of the simulation - that a good policy unroll will not deviate much from the ground truth, and adhere to regions where our simulation is valid. Note that we could easily modify this, by loading data at each time frame. This way, all data inputs become leaves in the graph, but we can still differentiate through the full unroll. In fact, we have experimented with this, but did not observe any improvements.
> >
> >
> > > 7. The related works need more recent methods to cover the whole problem including other end-to-end methods.
> >
> > We thank you for this suggestion and have extended our related work section accordingly, in particular adding a subsection on Neural Motion Planners.
> >
> >
> > > 8. The difference between the presented simulator and the differentiable simulator presented in [1] needs to be highlighted.
> >
> > The simulator proposed in our work is evaluating the position of the ego agent in closed-loop, while the position of other agents are used from the logs. This is a much simpler task than predicting trajectories of all agents, as in [1]. Our work demonstrates that such a simplified simulator is enough to learn meaningful policies, capable of being deployed in the real world. This comparison was added to the related work section.
> >
> > > 9. Line 165-167, "To remove this bias we discard the first K timesteps ...", an experiment showing a comparison between with or without discarding K timesteps is helpful.
> >
> > Please refer to Appendix B (Figure 2) for this.
> >
> > > 10. The introduction discusses covariant shifts between training and test distributions, but it is not clear how the proposed approach has addressed this problem.
> >
> > We added an explanation concerning this in the related work section, and thank you for pointing this out. In short, Ross et al. introduce the notion of covariate shift for the problem of Imitation Learning, proving a quadratic regret bound for naive models [2]. Later, they show that allowing interactive queries of an expert yields the optimal linear regret bound [3]. [4] prove this still holds when requiring less strict demands on the availability of the expert, and introduce the notion of synthetic corrections. As our approach is motivated by this and uses the same concept of corrections and expert queries (in each step, a synthetic correction to the expert’s ground truth position is used as a training signal), the same performance guarantees apply for us.
> >
> >
> > -----------------------------
> > [1] “TrafficSim: Learning to Simulate Realistic Multi-Agent Behaviors”, Suo, Regalado, Casas, Urtasun. CVPR 2021
> >
> > [2] “Efficient Reductions for Imitation Learning”, Ross, Bagnell, 2010
> >
> > [3] “A Reduction of Imitation Learning and Structured Prediction to No-Regret Online Learning”, Ross, Gordon, Bagnell, 2010
> >
> > [4] “Improving Multi-step Prediction of Learned Time Series Models”, Venkatraman, Hebert, Bagnell, 2015

---

> > > ### Comment · Reviewer_KT1F · 2021-09-04
> > > **Response to Authors**
> > >
> > > Thanks for taking the time and effort in responding to my comments and elaborating on your approach. I believe the paper has improved compared to the original version and the main concerns are adequately addressed in the new version and the updated supplementary material. The global response and response to my comments are convincing for the most part. I have updated my rating to weak accept.

---

### Official Review · Reviewer_wo2G · 2021-07-25

**Originality:** Good
**Technical Quality:** Fair
**Clarity Of Presentation:** Good
**Impact:** 3

**Recommendation:**

Weak Accept: I recommend accepting the paper, but will not argue for my recommendation if the majority of other reviewers have a different opinion.

**Summary:**

The present work proposes a novel differentiable data-driven scenario simulation technique that generates driving scenarios based on real-world data. This simulator is incorporated as a differentiable module in a policy gradient-based driving policy learning task. The authors evaluate the resulting policy learning system in comparison to several policy learning baselines on the Lyft Motion Prediction Dataset.

**Issues:**

__Main Paper__
- l. 79 / 80 What do you mean by "sim-to-real transfer" here? E.g. in the case of [23], the policy is not even retrained for the actual transfer.
- l. 129 & eq (2): In the current form of eq. (2), when the original and updated pose are the same, it seems like the new position becomes zero. Unless I am misunderstanding something, this does not look right.
- l. 138 do I understand correctly, that the simulator mainly allows for local perturbations of the expert trajectories rather then completely new scenarios in complex scenes such as intersections?
- l. 163: It is not clear to me why bias is introduced if the initial state of the trajectory is sampled from the expert? What characteristics of the expert driver's initial position are biased? Maybe something like speed?
- l. 198.: Could you provide implementation details on how exactly ChauffeurNet was modified (i.e. more details on your vectorization process for ChauffeurNet).

__Appendix__
- l. 16 How is the type embedding added? Simply via a one-hot representation?

**Reviewer Expertise:**

Good: General knowledge of the area

**Strengths And Weaknesses:**

__What I learned__
- I learned how differentiable driving scenario generators can be integrated into a policy learning pipeline.

__Strengths__
- Data-driven simulation engine improves the data efficiency of the training process.
- Differentiability of the simulator allows its use as a neural network module in a more complex fully differentiable pipeline.

__Weaknesses__
- The positioning and motivation of the work is not fully clear, are you focussing more on the simulator or the policy learning aspect? On the policy learning side, what is the ultimate goal of the paper? Mainly basic research in embodied AI for mobility? Or actual deployment of learned driving policies for Self-Driving Vehicles (SDVs)?
- Depending on the answers to the former, different (emphasis of the) evaluations may be necessary. For example, focusing on improved navigation behavior of real-world SDVs might require comparisons with methods based on trajectory-based optimization or other AV planning techniques such as Multi-Policy Decision Making (Cunningham et al., 2015). Another evaluation-related issue is that it is done purely on a dataset rather than on any actual driving platform or at least in some video-game environment.
- Finally, the simulator itself is not compared with other scenario simulators such as TrafficSim.
- Some implementation details are missing, for example how was the data split (did you simply use the default split given in the dataset?).
-  The code is unfortunately not available at the time of submission. I still think it is great that the authors plan on releasing it.
- At times some smaller aspects are hard to understand (some examples are given as questions in the issues section).

**Summary Of Recommendation:**

Overall, I see this work as going in a very interesting direction. My current objections are mainly motivation and evaluation related: The authors should also better clarify the motivation and positioning of the work. And the work needs to also evaluate the simulator in comparison to state-of-the-art. While the latter point may require significant work, I believe the motivation-related issues to be easily resolvable.

---

> ### Author Response · Authors · 2021-08-27
> **Addressing the raised issues**
>
> Dear Reviewer,
>
> Thank you very much for your helpful review and detailed questions. We would kindly ask you to first read our general comments in the answer to the AC above, and in this comment address your points in particular.
>
> > Some implementation details are missing, for example how was the data split (did you simply use the default split given in the dataset?).
>
> For training, we use the public train dataset “train.tar”, whereas for testing we use a random subset of the public “validate.tar”, which is roughly ¼ of the original size (25 h). This was done purely for the sake of faster experimentation and running more evaluations, and we argue that the used size of the evaluation dataset is large enough to guarantee statistical significance. We included these details in the amended version of our work.
>
> > l. 79 / 80 What do you mean by "sim-to-real transfer" here? E.g. in the case of [23], the policy is not even retrained for the actual transfer.
>
> Thank you for pointing out this paragraph, it was not written precisely. It is now updated, underlining the main difference between our method and the previously demonstrated sim-to-real transfer. In particular, these cited works did not show results when interacting with other agents.
>
> > l. 129 & eq (2): In the current form of eq. (2), when the original and updated pose are the same, it seems like the new position becomes zero. Unless I am misunderstanding something, this does not look right.
>
> We thank you for pointing this out and admit a slight abuse of notation. In some parts of the paper by $p_i$ we denote poses represented by their 3D coordinates (x, y, yaw) - whereas mainly we think of poses as members of $SE_2$, which is a group allowing rotation and translation. In this understanding, poses are represented by a rotation and translation - and when subtracting two identical poses, we do not obtain a 0-vector, but instead the identity element (which can be thought of as the identity matrix).
>
> > l. 138 do I understand correctly, that the simulator mainly allows for local perturbations of the expert trajectories rather then completely new scenarios in complex scenes such as intersections?
>
> That is correct.
>
> > l. 163: It is not clear to me why bias is introduced if the initial state of the trajectory is sampled from the expert? What characteristics of the expert driver's initial position are biased? Maybe something like speed?
>
> We thank you for this interesting question. In general, bias here is introduced by sampling from a different, biased distribution instead of the one we are interested in and optimize over in Eq (3): to be precise, we would like to optimize performance of our model, i.e. freely unroll our model (sample from this distribution) and compare these unrolls to the underlying expert samples. However, the starting point of such unrolls always lies within the expert distribution, thus the distribution used for sampling would be severely biased towards the expert one. This is a common problem in fields such as RL. There are several ways to account for this, such as Importance Sampling, or, what is used here, changing the sampling procedure to simulate different distributions. Your question though seems to be targeted at specifics of this bias, which is a very interesting but tough to answer question, as the distributions we are working with are very complex. Mainly though, the expert distribution will have a smaller variance w.r.t to x/y position around the target lane, as well as exhibit yaw values more closely aligning with the direction of the lane to follow. Furthermore, speed and other characteristics, yes, could also allow differentiating between the expert and our model distribution.
>
> > l. 198.: Could you provide implementation details on how exactly ChauffeurNet was modified (i.e. more details on your vectorization process for ChauffeurNet).
>
> For fairness, our re-implementation of ChaufferNet and our proposed method use the same backbone network and the same vectorization process. We present this process in detail in Appendix C: Policy architecture and state representation.
>
> > l. 16 How is the type embedding added? Simply via a one-hot representation?
>
> We do not use a one-hot representation and concatenate this with our original feature, but instead, directly add the type embedding to the vector. This is common practise when working with Transformers, compare e.g. [1] or [2] (in the latter a sinusoidal position embedding is added to the vectors in question, but the principle remains the same). Specifically, in our case the embedding is a learned set of vectors, which is added to the keys: in particular, initially each type is assigned a random type embedding vector, which is then refined over the course of training.
>
> [1] “End-to-End Object Detection with Transformers”, Carion, Massa, Synnaeve, Usunier, Kirillov, Zagoruyko, 2020
> [2] “Attention Is All You Need”, Vaswani, Shazeer, Parmar, Uszkoreit, Jones, Gomez, Kaiser, Polosukhin, 2017

---

> > ### Comment · Reviewer_wo2G · 2021-09-03
> > **Your Response**
> >
> > Thank you for responding to my review and providing further clarifications. With these clarifications and the additional experiments, I believe the work can potentially be published and I updated my rating accordingly.

---

### Official Review · Reviewer_q8Fh · 2021-07-25

**Originality:** Fair
**Technical Quality:** Good
**Clarity Of Presentation:** Good
**Impact:** 4

**Recommendation:**

Strong Accept: I recommend accepting the paper and will argue for my recommendation even if other reviewers hold a different opinion.

**Summary:**

The paper introduces a differential simulator for self-driving in urban areas and uses it to build an imitation learning approach.

**Issues:**

As think that the paper suffers from a limited comparison to the prior work. Also, the method for imitation learning introduced in the paper has limited novelty. However, I would strongly encourage the authors to continue their work since even benchmarking IL methods for self-driving and sharing their implementations is a valuable contribution on its own.

**Reviewer Expertise:**

Good: General knowledge of the area

**Strengths And Weaknesses:**

### Strengths
* The paper introduces a differentiable simulator for self-driving.
* The paper evaluates a method for imitation learning based on differentiation through the dynamics model of the simulator.
* The method is evaluated against several baselines.
* The authors are planning to share their open-source implementation for training and evaluation on the Lyft motion planning dataset.

### Weaknesses
* The paper lacks algorithmic novelty. Planning through a model has already been explored in prior work, for example, in Universal Planning Networks (Srinivas etal., 2018).
* The set of the baseline is not adequate. I would encourage the authors to compare with Adversarial Imitation Learning methods as well.

**Summary Of Recommendation:**

I would recommend the paper for acceptance solely for the reason that the authors promise to share their code. Having a good codebase to build further research on can be just as important for advancing the topic as introducing novel algorithms. After a quick googling, I wasn't able to find another repository that performs imitation learning on the Lyft dataset.

---

> ### Author Response · Authors · 2021-08-27
> **Open-sourcing and Universal Planning Networks.**
>
> Dear Reviewer,
>
> Thank you very much for your insightful review and for sharing the excitement about our work. We would kindly ask you to first read our general comments in the answer to the AC above, and in this comment address your points in particular.
>
> > “The set of the baseline is not adequate. I would encourage the authors to compare with Adversarial Imitation Learning methods as well.”
>
> We thank you for your suggestion. For Adversarial Imitation Learning methods, please refer to the global response. Another mentioned work - more concerning the novelty of our contribution - is Universal Planning Networks [1]. We share your excitement regarding Universal Planning Networks, but see large differences when comparing their method and ours, such as planning in a latent space.
>
> We are looking forward to sharing our code with you and the whole community - as mentioned above, it will be hosted on www.l5kit.org, the latest with the camera-ready version.
>
> Citations:
>
> [1] “Universal Planning Networks”, Srinivas, Jabr, Abbeel, Levine, Finn, 2018

---

### Author Response · Authors · 2021-08-31
**Global response**

*Posting the global response again, as it has disappeared from OpenReview.*

We thank the reviewers for their helpful comments. We structure the authors’ response threefold:

1. Summary of the response to the main points raised by the reviewers especially concerns related to the

a. experimental validation of the method,

b. nature of the novelty of our contribution.

2. Update of the paper submission; in particular, we added experimental validation of the method on real-world self-driving vehicles and clarifications on points raised by the reviewers.

3. Additional tailored responses to individual reviews and comments, reported in individual threads below each review.


### Part 1. Summary of the response to points expressed by at least 2 reviewers:

#### A. Concerns related to assumptions, experimental validation and benchmarking of the method.

Reviewers wo2G, KT1F, oEVn raised the concern that the assumptions taken by our method might limit its practical viability - in particular, use of the proposed simulator and resulting generalisation to real-world use-cases.

To resolve these concerns we conducted the additional experimental validation of deploying the method to control a real-world self-driving vehicle. This validation was carried out on a location with no overlap with the training data. As demonstrated, the vehicle was able to perform various maneuvers such as lane following, navigating intersections and keeping distance to other vehicles. This shows that the method can generalise well to real-world applications and new locations without the need of any fine-tuning or sim2real transfer. Details of this experiment are included in the paper submission and in the updated supplementary video. We believe, this demonstration provides important and substantial evidence of applicability of the presented method and constitutes one of very few works that were validated this way.

This also motivates our choice of baselines. Many imitation learning methods were demonstrated in simplistic simulated tasks (MuJoCo, CARLA…) but their applicability to complex, real-world applications - as well as techniques needed to scale - are unknown and can require significant engineering efforts to be realized. For this reason we decided to compare our work only to the methods that were demonstrated in real-world settings [1] (BC + perturbation) and [2] (BC), and not to ones that were not demonstrated this way, such as, GAIL [3] (suggested by reviewers q8Fh, oEVn). That’s being said, we anticipate that GAIL-style training can be interpreted as an extension of our work by training an extra discriminator network instead of using L2 imitation loss. As such we leave this for future work.

#### B. Concerns related to positioning and nature of the novelty of our contributions.

Both Reviewers q8Fh and oEVn raised that many of the components, such as use of simulation and policy learning, were demonstrated before and thus the positioning and novelty of the work is not clear.

We agree that much of the basic theory was invented before. These methods, however, were demonstrated only on simplistic simulated tasks, such as MujoCo, CARLA, etc. The task of urban self-driving is significantly more complex compared to these in terms of diversity (wide variety of common and outlier situations) and complexity (situations requiring complex reasoning and interactions no simulation can realistically capture). The exact degree to which these methods are applicable to tackle real-world problems, and techniques needed to do so, were not described and demonstrated before.

The novelty of our work lies in the combination of existing components to achieve previously impossible results. Specifically, we demonstrate that with recent approaches leveraging large datasets, mid-to-mid representations and data-driven simulation the problem of ML planning for self-driving is becoming tractable for real-world applications. This result is novel and potentially very impactful, especially considering the amount of effort and funding that was already invested towards solving real-world self-driving in the past decade.

We share the excitement with Reviewer q8Fh about open-sourcing our code and hope it will further enable the research community and industry to progress in this direction. We have received commitment from the authors of www.l5kit.org (the official repository of Lyft prediction dataset the method is evaluated on) to host the source code of our method and the baselines for easy access, benchmarking and future extensions. We pledge to make all this publicly available until the camera-ready version at the latest.

[1] “ChauffeurNet: Learning to Drive by Imitating the Best and Synthesizing the Worst”, Bansal, Krizhevsky, Ogale, 2018
[2] “Urban driving with conditional imitation learning” Hawke, Shen, et al, 2020.
[3] “Generative Adversarial Imitation Learning”, Ho, Ermon, 2016

---

> ### Author Response · Authors · 2021-08-31
> **Global response, continued**
>
> ### Part 2: Summary of changes to paper and supplementary video submission
>
> Section 2 - Related Work: We rewrote the comparison to sim2real methods [wo2G]. Further, we added additional methods - especially more recent end-to-end works - and added a paragraph detailing covariate shift and why our method mitigates this [KT1F].
>
> Section 5 - Results: We clarified the used data splits [wo2G]. Based on the recommendation by Reviewer KT1F, we re-ran all evaluations to include a new off-road error threshold of 2m, while moving the original table to the appendix. Please note that while doing so we discovered a wrong normalization factor in our previous results: previous reported I1K values were around 2x too high - however, this did not affect any of the relative comparisons or claims, as all methods were affected. We updated the reported values.
> We added Subsection 5.6, showing results of our proposed method being deployed on self-driving vehicles in the real world. This strongly argues effectiveness and real-world applicability, as well as answers questions regarding the generalization of our method - the test track the model was deployed on was not part of the training data. To make space for this, we moved the section containing results using auxiliary losses to the appendix.
>
> Appendix: In the appendix as well as supplementary video we now report and show results of our self-driving vehicle deployments. We further introduced Subsection 2.3, discussing used metrics and showing further results: in particular, we include here our original evaluation results with an off-road threshold of 4m, and break down ‘Comfort’ into the metrics (longitudinal) jerk and lateral acceleration, as recommended by Reviewer KT1F.

---

### Meta-Review · Area_Chair_J1yN · 2021-08-12

**Recommendation:** Accept (Poster)
**Confidence:** 5

**Metareview:**

The authors provided thorough responses that addressed the reviewers' concerns. Following the authors' responses, all reviewers are generally positive. The AC endorses the reviewers' recommendation to accept the submission.

---

> ### Author Response · Authors · 2021-08-27
> **Summary of the responses**
>
> We thank the reviewers for their helpful comments. We structure the authors’ response threefold:
> 1. Summary of the response to the main points raised by the reviewers, especially concerns related to the
>
> a. experimental validation of the method,
>
> b. nature of the novelty of our contribution.
>
> 2. Update of the paper submission; in particular, we added experimental validation of the method on real-world self-driving vehicles and clarifications on points raised by the reviewers.
> 3. Additional tailored responses to individual reviews and comments, reported in individual threads below each review.
>
> Part 1. Summary of the response to points expressed by at least 2 reviewers:
>
> A. Concerns related to assumptions, experimental validation and benchmarking of the method.
> Reviewers wo2G, KT1F, oEVn raised the concern that the assumptions taken by our method might limit its practical viability - in particular, use of the proposed simulator and resulting generalisation to real-world use-cases.
>
> To resolve these concerns we conducted the additional experimental validation of deploying the method to control a real-world self-driving vehicle. This validation was carried out on a location with no overlap with the training data. As demonstrated, the vehicle was able to perform various maneuvers such as lane following, navigating intersections and keeping distance to other vehicles. This shows that the method can generalise well to real-world applications and new locations without the need of any fine-tuning or sim2real transfer. Details of this experiment are included in the paper submission and in the updated supplementary video. We believe, this demonstration provides important and substantial evidence of applicability of the presented method and constitutes one of very few works that were validated this way.
>
> This also motivates our choice of baselines. Many imitation learning methods were demonstrated in simplistic simulated tasks (MuJoCo, CARLA…) but their applicability to complex, real-world applications - as well as techniques needed to scale - are unknown and can require significant engineering efforts to be realized. For this reason we decided to compare our work only to the methods that were demonstrated in real-world settings [1] (BC + perturbation) and [2] (BC), and not to ones that were not demonstrated this way, such as, GAIL [3] (suggested by reviewers q8Fh, oEVn). That’s being said, we anticipate that GAIL-style training can be interpreted as an extension of our work by training an extra discriminator network instead of using L2 imitation loss. As such we leave this for future work.
>
> B. Concerns related to positioning and nature of the novelty of our contributions.
> Both Reviewers q8Fh and oEVn raised that many of the components, such as use of simulation and policy learning, were demonstrated before and thus the positioning and novelty of the work is not clear.
>
> We agree that much of the basic theory was invented before. These methods, however, were demonstrated only on simplistic simulated tasks, such as MujoCo, CARLA, etc. The task of urban self-driving is significantly more complex compared to these in terms of diversity (wide variety of common and outlier situations) and complexity (situations requiring complex reasoning and interactions no simulation can realistically capture). The exact degree to which these methods are applicable to tackle real-world problems, and techniques needed to do so, were not described and demonstrated before.
>
> The novelty of our work lies in the combination of existing components to achieve previously impossible results. Specifically, we demonstrate that with recent approaches leveraging large datasets, mid-to-mid representations and data-driven simulation the problem of ML planning for self-driving is becoming tractable for real-world applications. This result is novel and potentially very impactful, especially considering the amount of effort and funding that was already invested towards solving real-world self-driving in the past decade.
>
> We share the excitement with Reviewer q8Fh about open-sourcing our code and hope it will further enable the research community and industry to progress in this direction. We have received commitment from the authors of www.l5kit.org (the official repository of Lyft prediction dataset the method is evaluated on) to host the source code of our method and the baselines for easy access, benchmarking and future extensions. We pledge to make all this publicly available until the camera-ready version at the latest.
>
> [1] “ChauffeurNet: Learning to Drive by Imitating the Best and Synthesizing the Worst”, Bansal, Krizhevsky, Ogale, 2018
> [2] “Urban driving with conditional imitation learning” Hawke, Shen, et al, 2020.
> [3] “Generative Adversarial Imitation Learning”, Ho, Ermon, 2016

---

> > ### Author Response · Authors · 2021-08-27
> > **Part 2: Summary of changes to paper and supplementary video submission**
> >
> > Section 2 - Related Work: We rewrote the comparison to sim2real methods [wo2G]. Further, we added additional methods - especially more recent end-to-end works - and added a paragraph detailing covariate shift and why our method mitigates this [KT1F].
> >
> > Section 5 - Results: We clarified the used data splits [wo2G]. Based on the recommendation by Reviewer KT1F, we re-ran all evaluations to include a new off-road error threshold of 2m, while moving the original table to the appendix. Please note that while doing so we discovered a wrong normalization factor in our previous results: previous reported I1K values were around 2x too high - however, this did not affect any of the relative comparisons or claims, as all methods were affected. We updated the reported values.
> > We added Subsection 5.6, showing results of our proposed method being deployed on self-driving vehicles in the real world. This strongly argues effectiveness and real-world applicability, as well as answers questions regarding the generalization of our method - the test track the model was deployed on was not part of the training data. To make space for this, we moved the section containing results using auxiliary losses to the appendix.
> >
> > Appendix: In the appendix as well as supplementary video we now report and show results of our self-driving vehicle deployments. We further introduced Subsection 2.3, discussing used metrics and showing further results: in particular, we include here our original evaluation results with an off-road threshold of 4m, and break down ‘Comfort’ into the metrics (longitudinal) jerk and lateral acceleration, as recommended by Reviewer KT1F.

---

### Decision · Program_Chairs · 2021-09-13

**Decision:**

Accept (Poster)

**Comment:**

The authors provided thorough responses that addressed the reviewers' concerns. Following the authors' responses, all reviewers are generally positive. The AC endorses the reviewers' recommendation to accept the submission.